Healthy-unhealthy animal detection using semi-supervised generative adversarial network

Almal Shubh 1
Bagepalli Apoorva Reddy 1
Dutta Prajjwal 2
http://orcid.org/0000-0003-1804-8590 Chaki Jyotismita 1 jyotismita@vit.ac.in
1 School of Computer Science and Engineering, Vellore Institute of Technology , Vellore, Tamil Nadu , India
2 School of Electronics Engineering, Vellore Institute of Technology , Vellore, Tamil Nadu , India
Moparthi Nageswara Rao
Electronic publication date: 2023 Feb 15
Publication date: 2023
Volume: 9
Electronic Location ID: e1250
Received 2022 Sep 15; Accepted 2023 Jan 20
Copyright: © 2023 Almal et al.
Copyright year: 2023
Copyright holder: Almal et al.
License: This is an open access article distributed under the terms of the Creative Commons Attribution License, which permits unrestricted use, distribution, reproduction and adaptation in any medium and for any purpose provided that it is properly attributed. For attribution, the original author(s), title, publication source (PeerJ Computer Science) and either DOI or URL of the article must be cited.
License URL: https://creativecommons.org/licenses/by/4.0/

Keywords: Healthy animal, Unhealthy animal, Semi-supervised Generative adversarial network, Fuzzy inference system, Deep learning

Funding: The authors received no funding for this work.

==============================
Background

Animal illness is a disturbance in an animal’s natural condition that disrupts or changes critical functions. Concern over animal illnesses stretches back to the earliest human interactions with animals and is mirrored in early religious and magical beliefs. Animals have long been recognized as disease carriers. Man has most likely been bitten, stung, kicked, and gored by animals for as long as he has been alive; also, early man fell ill or died after consuming the flesh of deceased animals. Man has recently learned that numerous invertebrates are capable of transferring disease-causing pathogens from man to man or from other vertebrates to man. These animals, which function as hosts, agents, and carriers of disease, play a significant role in the transmission and perpetuation of human sickness. Thus, there is a need to detect unhealthy animals from a whole group of animals.

Methods

In this study, a deep learning-based method is used to detect or separate out healthy-unhealthy animals. As the dataset contains a smaller number of images, an image augmentation-based method is used prior to feed the data in the deep learning network. Flipping, scale-up, sale-down and orientation is applied in the combination of one to four to increase the number of images as well as to make the system robust from these variations. One fuzzy-based brightness correction method is proposed to correct the brightness of the image. Lastly, semi-supervised generative adversarial network (SGAN) is used to detect the healthy-unhealthy animal images. As per our knowledge, this is the first article which is prepared to detect healthy-unhealthy animal images.

Results

The outcome of the method is tested on augmented COCO dataset and achieved 91% accuracy which is showing the efficacy of the method.

Conclusions

A novel two-fold animal healthy-unhealthy detection system is proposed in this study. The result gives 91.4% accuracy of the model and detects the health of the animals in the pictures accurately. Thus, the system improved the literature on healthy-unhealthy animal detection techniques. The proposed approach may effortlessly be utilized in many computer vision systems that could be confused by the existence of a healthy-unhealthy animal.

Introduction

Deep learning is an artificial intelligence subsystem that allows systems to progress and learn data driven problems without being coded. Deep learning is concerned with the development of computer programs that can access data and use it to understand for themselves. The learning process begins with analysis or data, such as examples, previous models, or suggestions, to make better decisions in the future. The main goal is to allow computers to train themselves without human intervention or assistance and to correct their own mistakes through this learning.

Animals suffer a lot of diseases, such as skin disease, infectious disease, and many more, which has to be taken care of by the respective owner, but some animals are left alone in the streets who are suffering from starvation, diseases, and harassment/abusing, such stray animals need to be identified and taken care of by animal shelters/animal care organizations (Zhuang & Zhang, 2019). It is difficult to devise an automatic labeling scheme for a pre-trained self-supervised model that applies to the classification task at hand. Transfer learning, on the other hand, has emerged as the preferred starting point for many computer vision applications. Consider a scenario in which a subset of our training data is labeled but the rest is not. Transfer learning is ineffective for the unlabeled portion of the training set. That is what the term “semi-supervised learning” refers to. This research project is based on classifying diseased animals who require utmost care and urgent medical assistance. This is the motivation for classifying unhealthy animals from healthy ones by using a semi-supervised learning technique. The classification model developed using semi-supervised GAN allows these organizations to get accurate details related to the animals. Modern Artificial Intelligence technologies enable the use of computer vision to distinguish ill animals from healthy ones, paving the way for the selection of unhealthy animals such as those who are weak, unwell, or mistreated.

Now-a-days deep learning is used in many sectors (Kumar et al., 2021; Aggarwal et al., 2022; Aggarwal, Dimri & Agarwal, 2020). As per our knowledge, this is the first article that will detect healthy and unhealthy animals. Automatic object recognition is without a doubt one of the most significant advances in computer vision, particularly in security systems, where convolutional neural networks (CNNs) are the current state-of-the-art solution. To achieve higher accuracies, recent CNN architectures have become deeper and wider (Hsu, Zhuang & Lee, 2020; Narin, Kaya & Pamuk, 2021). Even though CNNs have been widely used for photographic images, Haar, Elvira & Ochoa (2023) shows the effectiveness of CNNs as feature extractors for object recognition. However, the performance of CNNs mentioned above was based on evaluations of datasets that may not be representative of real-world conditions, particularly in the context of unhealthy animal detection. Some of the researchers used an Improved Feature Fusion Single Shot MultiBox Detector for the detection of sick broilers (Zhuang & Zhang, 2019). The main limitation of this study is the proposed method does not work very well if the broilers are too closely packed together. Some researchers used conditional GAN (cGAN) data augmentation to detect the defect in fruit (Bird et al., 2022). The main drawback of this approach is some time cGAN is unable to classify the data properly as it does not change the ground-truth likelihood and we are unable to predict the accuracy of the evaluated model’s density and declare that this image is dense enough to proceed with.

Some researchers have used some unlabeled images for checking the robustness of using GAN in image classifications. They have used an innovative technique called Positive-Unlabeled GAN (PUGAN) where real images are considered as positive and GAN generated images are negative. The generated data is treated as unlabeled data and made use of GAN for the classification purposes. Thus, they came to a conclusion that GAN can be implemented effectively for unlabeled image classification (Guo et al., 2020).

Some authors detailed their work to illustrate how GAN is used in semi-supervised learning. The research investigates the theoretical basis of GAN-based semi-supervised learning (GAN-SSL). Optimizing the discriminator of GAN-SSL for a given fixed generator is comparable to improving the discriminator of supervised learning. As a result, the top discriminator in GAN-SSL is expected to be error-free on labelled data. If the perfect discriminator can further lead the optimization aim to its theoretical maximum, the ideal generator will match the true data distribution. Because it is difficult to attain the theoretical maximum in practise, it is unrealistic to anticipate to construct a faultless generator for data generation, which appears to be the purpose of GANs. Furthermore, the authors predict that the best discriminator in GAN-SSL will be perfect on unlabeled data if the labelled data can traverse all connected subdomains of the data manifold, which is feasible in semi-supervised classification. Finally, by unexpectedly learning an imperfect generator, the minimax optimization in GAN-SSL has the potential to produce a perfect discriminator on both labelled and unlabeled data, i.e., GAN-SSL may successfully increase the discriminator’s generalization capacity by utilizing unlabeled information (Liu & Xiang, 2020).

Semi-supervised approaches have gained popularity as a result of recent developments in deep learning. One current technique to semi-supervised learning is generative adversarial networks (GANs) (SSL). A study describes a survey approach for SSL that employs GANs. Pseudo-labelling/classification, encoder-based, Triple GAN-based, two GAN, manifold regularization, and stacking discriminator techniques have all been used in the past to apply GANs to SSL. The various techniques are analyzed quantitatively and qualitatively. The R3-CGAN design has been recognized as the GAN architecture with cutting-edge outcomes. Given the recent success of non-GAN-based SSL techniques, future research potential for adapting SSL parts into GAN-based implementations are also suggested (Sajun & Zualkernan, 2022).

In some articles, authors have worked on image classification using GAN. In that case they used the Oxford-IIIT Pets dataset. It comprises of 37 different breeds of cats and dogs, each with a different scale, stance, and lighting, adding to the difficulty of the categorization process. In addition, we improved the performance of the recently developed GAN, StyleGAN2-ADA model to create more realistic images while avoiding overfitting to the training set. The authors accomplished this by training a modified version of MobileNetV2 to anticipate animal face landmarks and then cropping photographs accordingly. Finally, the authors integrated the synthetic images with the original data and compared their proposed technique to traditional GANs augmentation and no augmentation with various training data subsets. The authors confirmed their findings by assessing the precision of fine-grained image categorization using the most recent vision transformer (ViT) Model (Darvish, Pouramini & Bahador, 2022).

Despite the promising results obtained by GANs on paired/unpaired image-to-image translation, previous work frequently only transfers low-level information (e.g., texture or colour changes) and fails to manipulate high-level semantic meanings (e.g., geometric structure or content) of different object regions. While some researchers can generate appealing real images given a class name or description, they cannot condition on arbitrary shapes or structures, significantly restricting their application scenarios and model findings’ interpretation skills.

Recently, the GAN was effectively expanded to address semi-supervised picture classification problems. However, when tagged photos are few, it remains a significant barrier for GAN to use unlabeled images to improve its classification capacity. Wang, Sun & Wang (2022) present a novel CCS-GAN model for semi-supervised image classification, with the goal of improving classification ability by using the cluster structure of unlabeled photos and poorly produced images. It uses a novel cluster consistency loss to limit its classifier to maintain local discriminative consistency in each cluster of unlabeled pictures, providing implicit supervised information to enhance the classifier. Meanwhile, it employs an improved feature matching strategy to encourage its generator to generate adversarial images from low-density portions of the actual distribution, which can improve the classifier’s discriminative capacity during adversarial training and suppress the mode collapse problem. Extensive tests on four benchmark datasets reveal that the proposed CCS-GAN outperforms numerous state-of-the-art rivals in semi-supervised image classification tasks.

The different kinds of disease of tomato plant could also be detected via using GAN by some authors. Plant diseases and venomous insects pose a significant hazard to agriculture. As a result, early detection and diagnosis of many disorders is critical. The continual development of deeper deep learning methods has tremendously aided in the diagnosis of plant diseases, providing a robust tool with incredibly exact results; nonetheless, the accuracy of deep learning models is dependent on the volume and quality of labelled data used for training. In one of the studies (Abbas et al., 2021), the authors present a deep learning-based technique for detecting tomato illness using the conditional generative adversarial network (C-GAN) to produce synthetic images of tomato plant leaves. Following that, a DenseNet121 model is trained on synthetic and real pictures using transfer learning to classify tomato leaf images into ten disease categories. The proposed model has been thoroughly trained and tested using the publicly accessible PlantVillage dataset.

Based on the aforementioned findings, and to eliminate the reasoning procedures, a deep learning strategy-based network, semi-supervised generative adversarial network (SGAN), is proposed in this article to identify or distinguish between healthy and unhealthy animals in two folds. A strategy based on image augmentation is utilized before supplying the data to the deep learning network because the dataset has fewer images than usual. A combination of four different image augmentation techniques was used to increase the dataset along with the brightness correction technique. The penultimate step is to utilize a semi-supervised generative adversarial network (SGAN) to identify photos of healthy and ill animals.

The summarization of two folds of SGAN and the main contribution of this article is as follows: Creation of the augmented dataset: Data augmentation can help improve the performance and outcomes of machine learning models. The data augmentation tools enrich and supplement the data, allowing the model to perform better and more accurately. By introducing transformation into the datasets, data augmentation techniques reduce operational costs. All images in the dataset have four sorts of geometric changes done to them. Horizontal flipping, vertical flipping, rotation, scale-up, scale-down, and a combination of one to four are the variations. The combination of scale-up and scale-down is ignored since the ultimate outcome is reliant on the quantity of either scale-down or scale-up. For brightness correction, a fuzzy inference system is created.

Use of SGAN for healthy-unhealthy animal detection: The difficult problem of training a classifier in a dataset with a small number of labeled examples and a much larger number of unlabeled examples is known as semi-supervised learning. The generative adversarial network (GAN) is an architecture that uses large, unlabeled datasets to train an image generator model via an image discriminator model. The discriminator model can be utilized as a base for creating a classifier model. The SGAN, model is an extension of the GAN architecture that involves training a supervised discriminator, an unsupervised discriminator, and a generator model all at the same time. The result is a supervised classification model that extrapolates well to previously unseen instances, as well as a generator model that generates feasible instances. In this study, the discriminator is implemented with a non-pre-trained RestNet-18 model. With help of external GPU resources, the model application is enhanced.

This work goal is to further enable real-world usage by improving generalization with SGAN. The purpose of the research is to explore the use of SGAN to strengthen processes used in the detection of unhealthy animals, especially diseased ones.

The rest of the article is organized as follows: Section 2 describes the proposed approach, experimentations and results are reported in Section 3; Section 4 includes the analysis of the proposed approach with the approach used in the literature, and Section 5 concludes the article.

Materials and Methods

A broad overview of the proposed methodology is shown in Fig. 1.

Figure 1 Block diagram of the proposed system.

Pre-processing

Because of the computational difficulty of the methods involved, the pictures are reduced to 256 × 256 pixels; this resolution still enables the visualization of undesired aspects while balancing the total amount of RGB pixels. Because full-resolution photos are not practical for consumer-grade hardware, limiting the balance of the original model data decreases the quantity of memory needed to train all models. In terms of implementation and actual usage, sensors will have energy constraints due to the computation complexity and profit tradeoff associated with animal identification automation. As a result, this image size balance improves the approach’s usability. The black backdrop is substituted with white to improve noise detection throughout the generative process of learning. Though it does not influence the model’s training process, the backdrop is modified so that visual flaws may be better recognized through manual inspection during training.

Offline image augmentation has been applied to broaden the dataset and strengthen the model. The number of images in the dataset can be increased using this approach. The variability of the data is regularized, and removes the possibility of overfitting the model. The problem of scarce data causes a limitation in achieving generalized model which can give high training accuracy and have tremendous gap with test accuracy. Also, to ensure the number of training samples offline based method is used. All images in the collection have four sorts of geometric changes done to them. Horizontal flipping, vertical flipping, rotation, scale-up, scale-down, and a combination of one to four are the variables. The combination of scale-up and scale-down is ignored since the ultimate result is determined by the quantity of either scale-up or scale-down. The rotation, scale-down, and scale-up values are picked at random. The rotation span is set to 0° to 360° at the 2° separation, while the scale-down and scale-up factors are set to 1.0 to 0.5 and 1.0 to 2.0 at the 0.1 separations, respectively. The brightness of the figure is maintained by using fuzzy logic.

The goal of this pre-processing unit is to create a new fuzzy inference system (FIS) to improve the brightness/contrast of the animal image that resolves the limitations of conventional methods. FIS is created that takes the discrete pixel intensity, fuzzifies it by utilizing input membership functions (MFs), and then maps the input to the output using IF-THEN rules. Finally, applying the output MFs, a defuzzified value is generated.

To improve accuracy, the system employs seven input membership functions, as well as Gaussian and trap membership functions, which describe sets of pixel intensity values as linguistic variables such as very dark, dark, medium, bright, and very bright, as illustrated in Fig. 2. After evaluating several images, these membership functions were picked, and the range of pixel values for these linguistic variables was determined.

Figure 2 Input and output MFs.

The output MFs specify how much the pixel intensity should be raised or lowered depending on the knowledge base rules. Defuzzification is used to extract the crisp intensity modification value from the output MFs. Figure 2 depicts the output MFs. The core intuitive principle behind contrast enhancement is that if a pixel is dark, it should be darker, and if a pixel is bright, it should be brighter. An intuitive FIS is created around this concept. The algorithm employs the Mamdani FIS. The FIS includes an expert-created knowledge base that includes IF-THEN rules. These rules bridge fuzzy inputs to fuzzy outputs using a composite rule of intuition.

The suggested algorithm’s rules are as follows. IF the input is Very Dark, THEN the result will be Slightly Dark

IF the input is Dark THEN the result will be is Very Dark

IF the input is Medium THEN the result will be Slightly Bright

IF the input is Bright THEN the result will be Slightly Bright

IF the input is Very Bright THEN the result will be No Change

Peak signal to noise ratio (PSNR) is used to evaluate the performance of the suggested fuzzy logic-based image enhancement approach, and it is computed using Eq. (1).

(1) PSNR=10log10(G2RMSE)RMSE=∑M,N(RGB(M,N)−RGBE(M,N))2M×N

where, G is the maximum image intensity, (M, N) Is the image size, and RGB and RGBE are the original and enhanced images.

The new dataset is saved accordingly for the training and testing of SGAN.

SGAN for classification

Semi-supervised generative adversarial network (SGAN), generative adversarial networks are generative models that are built on a game theoretic situation in which a generator (G) network competes with a discriminator ( D) (Zhang, Pan & Zhang, 2021). The structure of SGAN used in this study is shown in Fig. 3.

Figure 3 Structure of SGAN.

The generator, given a noise variable Z as input, creates false samples with a distribution pg that matches the genuine data distribution p(data). The discriminator network, on the other hand, is taught to discriminate between actual samples (taken from training data) and false samples created by G. Generally, the discriminative model D is developed to increase its capacity to discriminate between genuine and false input data. By providing better false samples, the generator attempts to trick the discriminator. Quantitatively, the generator and discriminator engage in a two-player min-max game with the value function V(G,D)

(2) minmaxV(G,D)=Ex∼p(data)[log⁡(D(x))]+Ez∼pz[1−log⁡(G(z))]

where p(data) is the genuine data distribution, E is the expectation and pz is a noise probability, D(x) and G(z) are the discriminator and generator functions. Training a generative adversarial network may be thought of as an optimization procedure for both the generator and the discriminator. The generator’s output is denoted as pg.

The goal of SGANs is to reduce the Jensen–Shannon divergence between both the generative distribution pg and the data distribution p(data) with complete minimization achieved when pg =p(data). The optimum procedures for generator and discriminator are as follows:

By adding additional output in the discriminator, GANs may be generalized to semi-supervised learning. The discriminator’s first output just classifies data as true or false, but the second output categorizes data by class. The concept is that regardless of whether the data is real or fabricated, the classifier must assess if it can be categorized into the correct classes. If it can, the data is very likely accurate.

The classifier applied the Resnet-18 pre-trained model which consists of 18 convolutional layers. The generator model consists of five convolutional layers with batch normalization and ReLU activation functions included in each layer, Tanh function has been applied in the last layer.

Training phase

The training phase of the discriminator and generator is discussed in this section.

Train the discriminator (supervised)

Select a mini-batch of labeled real-world instances at random (x,y).

For the given mini-batch, compute D((x,y)) and backpropagate the multi-label loss to adjust θ(D) to minimize the loss.

Update the discriminator by ascending its stochastic gradient ( m is the number of training samples):

(3) ∇θd1m∑[logD(x(i))+log(1−D(G(z(i))))]

Train the discriminator (unsupervised)

Select a mini-batch of unlabeled real-world instances at random x.

For the provided mini-batch, compute D(x) and backpropagate the classification model loss to adjust θ(D) to minimize the loss.

Create a mini-batch of false samples from a mini-batch of random noise vectors z: G(z)=x′.

For the provided mini-batch, compute D(x) and backpropagate the classification model loss to update θ(D) to minimize the loss.

Update the discriminator by ascending its stochastic gradient:

(4) ∇θd1m∑[logD(x(i))+log(1−D(G(z(i))))]

The discriminator network makes use of the fine-tuned transfer learning idea from a big ImageNet-trained model, ResNet-18.

A linear search of neurons is used to optimize the number of neurons in the interpretation layer. The network is allowed a max of 100 epochs to train, however, training is terminated after 10 epochs if no additional learning happens. The discriminator is trained on unlabeled data, then on labeled data, and finally on produced data, with the loss being measured after each training dataset.

(5) L(D)=L(D_supervised)+L(D_unsupervised)

(6) L(D_supervised)=−Ex,yp(data)(x,y)logpD(y|x,y<K+1

(7) L(D_unsupervised)=−Exp(data)(x)log[1−pD(y=K+1|x)]−Ezp~z(z)pD(y=K+1|G(z))

where k is the number of classes and pD is the probability of data.

Train the generator

Mini-batch of fictitious instances can be created by using a mini-batch of random noise vectors, z, as follows: G(z)=x.

For the given mini-batch, compute D(x) and backpropagate the classification model loss to update θ(G) to maximize the loss.

Update the generator by descending its stochastic gradient:

(8) ∇θ=d1m∑[log(1−D(G(z(i))))]

In the generator architecture, the number of channels, the size of the z latent vector, and the size of the feature maps are all initialized. The initial input to the generator is a vector representing a three-channel 256 × 256-pixel image (256 × 256 × 3) inputting random vector z in the first 7 × 7 convolutional layers out of the four convolutional layers, followed by 14 × 14 convolutional layer, 28 × 28 convolutional layer, and 112 × 112 convolutional layer with Batch Normalization to enhance stable neural network and ReLU activation function was chosen. The Generator model’s output layer is built using a hyperbolic tangent activation layer for scaling, and the Adam optimizer is being used to train it. The generator loss is calculated as follows.

(9) L(G)=∥Exp(data)(x)f(x)−Ezpz(z)f(G(z))∥22

Results

For the training of SGAN, Intel Core i7, 2.6 GHz 6-Core processor, NVIDIA GeForce GTX 1600 Ti Graphics (6 GB) and 16 GB 2667 MHz DDR4 RAM, are used which minimizes the time of training. TensorFlow 2.9.2 is used for the suggested approach’s implementation.

Dataset

Initially, an open-source dataset of images of cats and dogs was obtained from common objects in context (Lin et al., 2014). The dataset comprises 1,500 photos of cats and dogs that are labeled in COCO format and have a resolution of roughly 250 × 250 pixels. Given that each COCO annotation specifies a single class, i.e., an animal with skin illnesses discovered by images, we filter through the dataset to assign a single binary class label of “healthy” or “unhealthy” to each animal image. As the dataset has few numbers of images, image augmentation is applied as mentioned in “pre-processing”. Ultimately 30,000 images are used for the study. Among that 15,000 are from dog images and the remaining are from cat images. Among 15,000, 7,000 are from healthy animals and the rest of the images are from the unhealthy animal image. A total of 60% of the images are used for training purposes, rest of the images are used for texting purposes. Figures 4 and 5 show some sample dataset images.

Figure 4 Healthy animal samples.

Figure 5 Unhealthy animal samples.

Pre-processing

Data augmentation was done in this stage and saved the images. Scale up, scale down, flip, and orientation are randomly applied in a combination of one to four to the images and saved to the dataset. After pre-processing the number of images raised to 3,468. The suggested fuzzy image brightness improvement technique’s output is compared to the output of two existing well-known image enhancement methods: contrast limited adaptive histogram equalization (CLAHE) and classic histogram equalization (HE). Figure 6 compares the results of CLAHE, HE, and the suggested fuzzy image brightness enhancement approaches.

Figure 6 (from left) Original image, brightness transformation using the proposed approach, brightness transformation using CLAHE, Brightness transformation using HE.

PSNR is used to assess the performance of the suggested fuzzy image brightness enhancement technology. The average PSNR achieved by utilizing CLAHE, HE, and the suggested fuzzy image brightness enhancement approach are 78.03, 77.01, and 83.29, respectively.

Performance evaluation of SGAN

Table 1 shows the results obtained for the healthy and unhealthy animal image classification data set using the original dataset.

Table 1 Performance evaluation using original dataset.

Evaluation metrics	Score (%)	
Accuracy	36	
Precision	36	
Recall	34	
F1 score	35	
Specificity	33	
AUC-ROC	36	

After adding augmented images scored higher than training only on the real images. It is been observed that we achieved around 91% of accuracy (see Table 2), and we obtained better results after training for 200 epochs. Figures 7–10 shows the ROC curve for the true positive rate and false positive rate.

Table 2 Performance evaluation of the proposed method.

Evaluation metrics	Score	
Accuracy	0.91	
Precision	0.907	
Recall	0.883	
Specificity	0.870	
AUC-ROC	0.9	
F1-score	0.895	

Figure 7 ROC curve.

Figure 8 ROC curve.

Figure 9 ROC curve.

Figure 10 ROC curve.

Some experimentations are done to check whether the system is cost efficient or not. Figure 11 demonstrates the cost function vs epoch plot. From Fig. 11 it is cleat that the method is cost efficient as the cost is decreasing while increasing the epoch.

Figure 11 Cost function vs epoch plot.

The analysis of the proposed method is done by comparing the method with the existing one. The augmented dataset is used for this purpose. Table 3 shows the comparison of the same.

Table 3 Comparison with other methods in the literature.

Model	Accuracy	Precision	Recall	Specificity	AUC	F1 score	
Conditional GAN (Kumar et al., 2021)	0.774	0.79	0.86	0.79	0.77	0.81	
Deep convolutional generative adversarial network (DCGAN) (Basavaiah & Arlene Anthony, 2020; Wu, Chen & Meng, 2020)	0.872	0.86	0.862	0.86	0.85	0.85	
Auxiliary classifier generative adversarial network (ACGAN) (Chaabane et al., 2022; Saha & Sheikh, 2020)	0.887	0.87	0.88	0.86	0.851	0.874	
Decision tree classifier (Reshi et al., 2021)	0.788	0.749	0.759	0.764	0.769	0.733	
Support vector machine classifier (SVM) (Lin et al., 2021)	0.868	0.849	0.85	0.88	0.859	0.848	
Convolutional neural network (CNN) (Patel & Upla, 2022)	0.848	0.84	0.845	0.83	0.82	0.842	
Graph neural network (GNN) (Guo et al., 2020)	0.83	0.82	0.825	0.81	0.829	0.822	
Autoencoder (Patel & Upla, 2022)	0.88	0.875	0.878	0.86	0.88	0.872	
The proposed method (SGAN)	0.914	0.907	0.883	0.893	0.90	0.895	

From Table 3, it is clear that the proposed method can achieve the highest accuracy among other methods used in this analysis. Here some possible reasons are discussed. With conditional GAN, guaranteeing the detectability of adversarial-perturbed inputs is impossible. One of the main drawbacks of the decision tree is it is largely unstable compared to other decision predictors. It is less effective in predicting the outcome of a continuous variable. The SVM algorithm is not suitable for large data sets. SVM does not perform very well when the data set has more noise i.e., target classes are overlapping. In cases where the number of features for each data point exceeds the number of training data samples, the SVM will underperform. Also, it is incapable to extract features effectively. Since convolutional neural networks are typically used for image classification, we are generally dealing with high-dimensional data (images). While the structure of ConvNet aims to mitigate over-fitting, you generally need a large amount of data for a convolutional neural network to work effectively. Of course, the amount of data you need depends on the complexity of the task at hand. GNNs are not robust to noise in graph data. Adding a slight noise in the graph through node perturbation or edge addition/deletion is having an adversarial effect on the output of the GNNs. The main limitation of the autoencoder is while training the autoencoder a lot of data, processing time, hyperparameter tuning, and model validation are needed before we even start building the real model. An autoencoder learns to efficiently represent a manifold on which the training data lies. If the training data is not representative of the testing data, then one can wind up obscuring information rather than clarifying it. An autoencoder learns to capture as much information as possible rather than as much relevant information as possible. The performance evaluation of the proposed SGAN, ACGAN, and DCGAN is comparable to each other. SAGN achieved better results among the three, therefore, we used SGAN for the detection of healthy, unhealthy animals.

Semi-supervised GAN was implemented for the classification of images. The dataset consists of a total of 3,468 images. A fuzzy inference system is used to correct the brightness of the image. With the increase in epochs the system initially gave 41% accuracy, with change parameters that are learning rate and optimizers of Generator and Discriminator resulting in 46% accuracy. Dataset with augmented techniques such as rotation, flipping, and resize which finally resulted in 79%, with the change in optimizer value and adding the weight decay which successfully resulted in 91.4%. The other supervised or semi-supervised learning modes could not perform up to the mark. Also, different types of GANs could not give the level of accuracy that is given by SGAN. For example, the results of our model give a 91.4% accuracy whereas the conditional GAN (77.4% accuracy) or DC GAN (87.2% accuracy) gives less accuracy. The usage of AC GAN (88.7% accuracy) comes close to the accurate prediction but still less than SGAN. So, comparing these models we can say our SGAN model is a more accurate fit for this particular application.

Conclusions and future scopes

A novel two-fold animal healthy-unhealthy detection system is proposed in this study. In these two folds, data augmentation and classification are done. In data augmentation, four different transformations are used orientation, scale-up, scale-down, and flip in the combination of one to four. This range of training images enables the system to accommodate differences in animal patterns in size, orientation, and flipping, resulting in improved generalization. A fuzzy inference system is created to enhance the brightness of the image. A fuzzy inference system is used to overcome the limitations of traditional brightness enhancement techniques. In the second fold, a semi-supervised GAN is used to detect the healthy-unhealthy animals. Here, the semi-supervised technique is used to overcome the limitation of a fully supervised technique. In a fully supervised technique, all labeled images are needed whereas semi-supervised GAN can handle both labeled and unlabeled images. The result gives 91.4% accuracy of the model and detects the health of the animals in the pictures accurately. Thus, the system improved the literature on healthy-unhealthy animal detection techniques. The proposed method is easily applicable to various computer vision systems that may be confused by the presence of a healthy-unhealthy animal.

In this article, we mainly focused on some datasets of healthy and unhealthy dogs and cats. In our future work, we can use the model for different animal or bird pictures to get the desired result with the same accuracy.

Supplemental Information

Supplemental Information 1 COCO dataset.

Click here for additional data file.

Additional Information and Declarations

Competing Interests

Author Contributions

Data Availability

Jyotismita Chaki is a Section Editor for PeerJ Computer Science

Shubh Almal conceived and designed the experiments, performed the experiments, analyzed the data, performed the computation work, prepared figures and/or tables, authored or reviewed drafts of the article, and approved the final draft.

Apoorva Reddy Bagepalli conceived and designed the experiments, performed the experiments, analyzed the data, performed the computation work, prepared figures and/or tables, authored or reviewed drafts of the article, and approved the final draft.

Prajjwal Dutta conceived and designed the experiments, performed the experiments, analyzed the data, performed the computation work, prepared figures and/or tables, authored or reviewed drafts of the article, and approved the final draft.

Jyotismita Chaki conceived and designed the experiments, performed the experiments, analyzed the data, prepared figures and/or tables, authored or reviewed drafts of the article, and approved the final draft.

The following information was supplied regarding data availability:

The code is available at GitHub and Zenodo: https://github.com/shubh0125/Healthy-Unhealthy-Animal-Classification.

Apoorva Reddy, & Shubh Almal. (2022). apoorvareddy612/Healthy-Unhealthy-Animal-Classification: Healthy-Unhealthy Animal Classification (v1.0.0). Zenodo. https://doi.org/10.5281/zenodo.7475604.

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
