# Peer review of "Healthy-unhealthy animal detection using semi-supervised generative adversarial network"

_PeerJ Computer Science, doi:10.7717/peerj-cs.1250_

## Round 0.1 · original submission · Major Revisions

Good Work Carried out . But we have some suggestions to make the article more technical strength.

Image Augmentation can be done online during training or offline before training starts. Authors preferred the offline approach, why?

What is the version of Tensorflow? Give the necessary information for the reproducibility of the proposed study.

The introduction part is not sufficient, literature is not presented sufficiently. Thus, the number of referenced studies is insufficient.

Based on reviewers' suggestions update and resubmit another version.

Reviewer 1 ·

Basic reporting

- Instead of using 'man' use 'mankind' or a similar proper word
- Line 49, instead of using 'involuntarily' prefer data-driven and refine the sentence accordingly
- Line 64, 'classifying diseased animals from healthy animals' refine this sentence since this part is not meaningful, i.e. remove 'healthy' word
- Introduction is not well organized
- A visual result for Fuzzy Inference System (FIS) should be given.
- all symbols should in in math style, i.e., 'pz' (noise probability) in the text
- Figure qualites (1, 2, 3, 7)are low. Use EMF for word and SVG/pdf for latex to create figures in vector format. Such figures provides very high quality for reading and printout. Use of these vector format also lead to reduced pdf size for the paper
- Figure 2 is not informative at all
- In table 5, highlight the best performance as bold. Also, consider presenting Table 5 as figure since tabular information is difficult to grasp by the reader

Experimental design

- Semi-supervised Generative Adversarial Network are already proposed in the literature. Also using data augmentation is not a novelty as well.
- Authors says that 'full-resolution photos are not practical for consumer-grade hardware' and 'sensors will have energy constraints' which is not a limitation of the method but limitation of the hardware they have or maybe they aim for developing an embedded system. If so, this should be clearly stated.
- Image Augmentation can be done online during training or offline before training starts. Authors preferred offline approach, why?
- What is the version of Tensorflow? Give necessary information for reproducibility of the proposed study.

Validity of the findings

- Did labeling of 1500 photos of cats and dogs are done by by experts?
- What kind of ilnesses have these cats and dogs? give statistics (frequency) of these ilnesses, i.e. 150 dogs have xxx ilness
- Did you use train-test set approach or train-validation-test approach? This is important to judge the real indication of the performance (91% of accuracy) given in the paper. If rain-validation-test approach is used then 91% of accuracy seems reasonable but if train-test set approach is used then 91% of accuracy is not difficult to obtain
- Authors suggest that SVM is not sufficient, and they list possible reasons. But one of the main reason is incapability of SVM to extract features automatically, unlike CNN.
- In figure 5, backgrounds are removed for some unhealty animals. Why? Is this common in the whole dataset?
- Figure 7, ROC curve, should be generated with more operating points. In its current form it is only generated for FPR=0, FPR=0.25, FPR=1, which causes curve and AUC to be inaccurate.

Additional comments

- Introduction part is not sufficient, literature is not presented sufficiently. Thus, number of referenced studies are insufficient.
- Authors should cite some studies from PeerJ Computer Science as they thought their study is contributing the studies in PeerJ. If they could not able to cite some studies from PeerJ then that means their study is not a proper one for this specific journal.

·

Basic reporting

comments are mentioned in a separate file.

Experimental design

comments are mentioned in a separate file.

Validity of the findings

comments are mentioned in a separate file.

---

## Round 0.2 · Minor Revisions

Dear Authors,

Your manuscript looks good, and you have made improvements. However, there are a few suggestions remaining. Please complete and re-submit for a final decision.

·

Basic reporting

All comments addressed by the authors

Experimental design

All comments addressed by the authors

Validity of the findings

All comments addressed by the authors

Additional comments

to strengthen the proposed work, the authors can include the following works
1. Semantic Information Extraction from Multi-Corpora Using Deep Learning
2. Solving User Priority in Cloud Computing Using Enhanced Optimization Algorithm in Workflow Scheduling
3. Statistical Performance Evaluation of Various Metaheuristic Scheduling Techniques for Cloud Environment

Reviewer 3 ·

Basic reporting

Clear and unambiguous, professional English used throughout.

Experimental design

Original primary research within Aims and Scope of the journal.

Validity of the findings

Impact and novelty not assessed. Meaningful replication encouraged where rationale & benefit to literature is clearly stated.

Additional comments

1. Is it cost-efficient?
2. Revise the references with the latest references that were published in 2020–2023, and update the literature accordingly.

---

## Round 0.3 · accepted · Accept

Dear Author,

Good work , you have completed all of the requirements of the reviewers.